# Scoping the Evolution of Corporate Social Responsibility (CSR) Research in the Sustainable Development Goals (SDGs) Era

**Amr ElAlfy** [1], **Nicholas Palaschuk** [1], **Dina El-Bassiouny** [2], **Jeffrey Wilson** [1,*] and **Olaf Weber** [1]

1   School of Environment, Enterprise, and Development (SEED), University of Waterloo, 200 University Ave W, Waterloo, ON N2L 3G1, Canada; aelalfy@uwaterloo.ca (A.E.); npwpalaschuk@uwaterloo.ca (N.P.); oweber@uwaterloo.ca (O.W.)
2   College of Business, Zayed University, Abu Dhabi P.O. Box 144534, UAE; dina.elbassiouny@zu.ac.ae
*   Correspondence: jeffrey.wilson@uwaterloo.ca; Tel.: +1-(519)-888-4567-x40049

**Abstract:** Amidst a contemporary culture of climate awareness, unprecedented levels of transparency and visibility are forcing industrial organizations to broaden their value chains and deepen the impacts of Corporate Social Responsibility (CSR) initiatives. While it may be common knowledge that the 2030 agenda cannot be achieved on a business-as-usual trajectory, this study seeks to determine to what ends the United Nations Sustainable Development Goals (SDGs) have impacted CSR research. Highlighting linkages and interdependencies between the SDGs and evolution of CSR practice, this paper analyzes a final sample of 56 relevant journal articles from the period 2015–2020. With the intent of bridging policy and practice, thematic coding analysis has supported the identification and interpretation of key emergent research themes. Using three descriptive categorical classifications (i.e., single-dimension, bi-combination of dimensions, sustainability dimension), the results of this paper provide an in-depth discussion into strategic community, company, consumer, investor, and employee foci. Furthermore, the analysis provides a timely and descriptive overview of how CSR research has approached the SDGs and which ones are being prioritized. By deepening the understanding of potential synergies between business strategy, global climate agendas and the common good, this paper contributes to an increased comprehension of how CSR and financial performance can be improved over the long-term.

**Keywords:** corporate social responsibility (CSR); sustainability; sustainable development goals (SDGs)

## 1. Introduction

In a paradigm characterized by unprecedented levels of transparency and visibility, public stakeholders and disclosure standards have gained considerable power in their ability to drive trends toward more sustainable business practices. Amidst the advent of the United Nations' Sustainable Development Goals (SDGs), global sustainability discourse has progressed to a point where it is inseparable from the role of the firm [1]. What must be considered a keystone element of progressive competitive strategies, creating shared value for the common good has become integral to Corporate Social Responsibility (CSR) in a way that changes the narrative on 'what' constitutes CSR and 'how' companies approach it in practice [2]. Under cognitive framings of managerial decision-making, past CSR behavior(s) and associated performance implications have been shown to strongly influence the perceptions of leadership regarding the relevance of social and environmental issues in value creation [3]. Conceptualized under ethical motives for societal well-being, the proliferation of business case(s) for CSR now materializes as a fiduciary duty and the sustainability case of business [4]. As the concept of CSR evolves, it is critical to understand how the SDGs and sustainability more

broadly are influencing corporate strategy, CSR agendas, reporting practices, disclosure mechanisms, stakeholder expectations, and regulatory requirements.

The motivations for investing in CSR initiatives and integrating them into business strategy are grounded in a shared desire to ensure a firm's long-term success and survival [5]. By aligning the purpose and values of CSR with market drivers and stakeholder demands, CSR practices have become due diligence for preserving the firm's license to operate, avoiding reputational damages, building loyalty, and maintaining competitive positioning [6]. Empirically grounded, the impacts of CSR on financial performance can be explained through top-line growth [7], decreased cost to capital, increased reputation and goodwill [8], and reduced technical and material risks [9].

Further, recent studies have shown that firms with well-coordinated and self-organized CSR strategies outperform their counterparts across similar industry groupings [10,11]. Superior share price performance has also been exhibited by companies listed on sustainability indices (i.e., *Dow* Jones Sustainability Index, *FTSE4Good*) when compared to companies listed in their non-sustainable counterparts [12]. While a notable rise in the number of companies publishing CSR reports can be observed, the quality and consistency of content being disclosed vary significantly [13]. This becomes further compounded by the heterogeneity amongst global reporting standards and a divergence in rating(s) criteria. According to Berg et al. (2019), this is what can be referred to as "*aggregate confusion*" [14]. Even with nearly 1400 companies, spanning 160 countries operating as signatories to the *United Nations Global Compact* [15], the simple fact remains that companies are afforded an overly flexible disclosure process that reinforces issues of evaluation, comparability, and ultimately usefulness [16].

Whether pursuing business cases of CSR is enough to satisfy global sustainable development remains subject to debate within and across academic disciplines. Often resting on *a priori* organizational frameworks, the legitimacy of this logic falls short when sustainable development is reduced under neo-liberal economic rationality or economic performance leveraged with coincidental CSR contributions [17]. In practice, bottom-line implications are left vulnerable to capricious public opinions, senior management turnover, and quarterly financial cycles [18]. Deeply ingrained throughout conventional cost accounting and performance management is a utilitarian view that rewards manager's and senior leadership when acting as self-seeking opportunistic individuals with the intent to maximize personal economic interests [19]. Materializing in the form of 'greenwashing', the reduction of CSR under win-win scenarios at the intersection of the triple bottom long constitutes a key managerial motivation for CSR and a conventional approach to building the business case [20]. Rather than an end in-of-itself, CSR activities are treated as philanthropic add-ons necessary for catering to current public opinion while securing loyalty [17,21]. This does little in the way of transforming organizational behaviour in a manner that is required to support meaningful progress on the SDGs. This underscores the fact that the very notion of 'doing well by doing good' is fundamentally a proposition of diminishing returns [22,23].

Demonstrable of a lack of managerial know-how and information for intervention selection/design, research indicates that such realities negatively mediate management's motivation/commitment to CSR [12]. Until this rationale is addressed systematically, strategic CSR literature will continue to turn out isolated success stories. As identified by Schaltegger et al. (2012), this will require that the formulation and implementation of strategy moves away from those that only strive for market sustainability through competitive advantages in the sense of the Resource-Based View (RBV) of the firm. By aligning the purpose and values of CSR with market drivers and stakeholder demands, CSR practices have become due diligence for preserving the firm's license to operate, avoiding reputational damages, building loyalty, and maintaining competitive positioning [9]. Empirically grounded, the impacts of CSR on financial performance can be explained through top-line growth [7], decreased cost to capital, increased reputation and goodwill [8], and reduced technical and material risks [9].

With respect to research, Bansal and Song [24] highlight the fact that, despite novel insights being made on the role of the firm and its embeddedness within the business-society-nature

interface, the variability among its subjective interpretations has limited construct validity in practice. Nevertheless, since the introduction of the SDGs, many firms have begun to strategically engage with the international framework as a means of creating functional linkages between performance outcomes and the common good [25]. An integrated framework comprised of 169 targets and 232 unique indicators, the SDGs have shifted CSR discourse from being reactive to stakeholders' mandates to a proactive one that helps firms play an active role in influencing sustainable development trajectories [26].

Therefore, this paper aims to provide a scoping review to synthesize academic literature on CSR since the global adoption of the SDGs. With the intent to deepen understandings of whether and how it has affected CSR, research articles were retrieved from the period 2015–2020. This paper seeks to deepen the understandings of (1) how global adoption of the SDGs has influenced academic literature on strategic CSR and (2) which new elements define CSR reporting best practices in the SDG era? Theoretically, this paper draws on strategic CSR literature [24,27] to provide a holistic perspective on 'how' and 'why' firms are integrating CSR into core planning, processes, and structures while attempting to create both social value and corporate value [28].

Using three scientific databases (i.e., Sci-verse Scopus, Science Direct, ISI Web of Science), the review conducted in this study consisted of a final sample of 56 peer-reviewed articles. By exploring 'whether' and 'how' the advent of the SDGs framework has impacted strategic CSR research, this paper (1) identifies trends in research output, (2) identifies key gaps within the existing literature, and (3) elaborates on current understandings of CSR-SDG linkages to identify opportunities and aid future research. The remainder of the paper is structured as follows: First, a historical review of the evolvement of the CSR literature is provided and the case for the theory of strategic CSR is positioned. Next, the method adopted to conduct the scoping review is disucssed and then the results of the study are shared. Following, several research implications are highlighted and avenues for future research are presented and, finally, the conclusions and research limitations of this study are provided.

## 1.1. Background

CSR has existed in the academic literature for more than fifty years without a global consensus on its definition or a standard set of application criteria [29]. The current reporting landscape covers a wide range of topics including social issues, philanthropy, sustainability, and environmental issues, and an ever-changing set of terminology to capture the ethos of the concept. While the underlying frameworks underpinning these abstractions may imply differing ideals of firm purpose, they share a normative belief that companies have a responsibility beyond pure profit-seeking that includes economic, social, and environmental concerns. The integration of these three dimensions explains the proliferation of the term 'sustainable' as a core concept in CSR nomenclature [30].

Furthermore, the adoption of the term 'sustainable/ility' introduces further definitional confusion. The question remains, is sustainability a dimension of CSR or does it imply an expansion of the concept beyond simply the 'social' aspect and require the use of a new lexicon? The interchangeability of open-ended terms like sustainability and responsibility as part of CSR literature has perpetuated ambiguity and has caused the meaning of the concept to be widely interpreted. Bansal and Song [24] highlight the systemic issues for both research and practice that arise when managers and academics alike use the words 'responsibility' and 'sustainably' interchangeably, inconsistently, and ambiguously. Porter and Kramer [31] highlight the fact that, while lauded conceptually, incongruencies throughout CSR measurement and lack of strategic orientation undermine corporate progress on sustainable development. This in part is addressed by Drucker (1984), who referred to the notion of strategic CSR, which can enhance the competitive advantage of corporations [27]. Under this paradigm, CSR is no longer seen as an add-on to business operations but as a strategy that is cascaded across all functions [32]. Werther and Chandler [28] define "Strategic CSR", as "the incorporation of a holistic CSR perspective within a firm's strategic planning and core separations so that the firm is managed in the interests of a broad set of stakeholders to achieve maximum economic and social value over

the medium to long-term" [28] (p. 40). Strategic CSR offers a new lens to underpin CSR focused on strategic and operational integration as a means of improving competitiveness, performance, and profitability [28,31].

An additional challenge is that, while CSR is a global concept, it is applied differently across social, economic, legal, and political contexts. As an inherent part of the CSR concept, this remains true for communicating and reporting on CSR engagement, which, by its very nature, is affected by differing regulatory requirements, disclosure mechanisms, and stakeholder expectations. Fifka [33] studied how research approaches regarding CSR reporting differ across countries and regions. Cultural and geographic heterogeneity from both norm-based and regulatory frameworks pose undeniable challenges. This inconsistency highlights the need for a globally accepted reporting framework and disclosure mechanisms. Before achieving these needs, the business community requires a shared vision to frame CSR. The UN Sustainable Development Goals (SDGs) offer that vision, and, more importantly, an opportunity to align business models with national commitments to sustainable development [25].

Common wisdom holds that sustainable development, across all levels, is not possible without the sustainable development of corporations [34]. With respect to CSR practice, sustainable development literature has become particularly relevant in envisioning development pathways, defining actionable goals, creating indicators, and asserting values [35]. Such is the paradigm of Corporate Social Responsibility [22]. Dependent on common-pool resources as systems inputs, these firms have a shared responsibility to contribute to societal well-being. By doing so, the needs of future generations become internalized in organizational culture and corporate value chains [24].

*1.2. Sustainable Development Goals (SDGs)*

The Sustainable Development Goals were adopted by the United Nations General Assembly in September 2015 as part of the 2030 Agenda for Sustainable Development. Characterized as a "new, universal set of goals to develop a global vision for sustainable development by balancing economic growth, social development, and environmental protection" [36]. The SDGs can be seen as a novel approach to global governance through goal-setting and tailored eco-feedback processes. The SDGs were developed through inter-governmental collaboration using public engagement processes to actively mobilize and consult national governments of both 'developing' and 'industrialized' countries in addition to various civil society groups [37].

As successors to the Millennium Development Goals (MDGs), the SDGs are expected to do better in addressing issues of sustainability. Moving well beyond the scope of the MDGs and the traditional "three-pillar" approach to sustainable development, the SDGs framework is intended to be universal, calling for integrative approaches that link human development and environmental sustainability [38]. As a policy framework guiding society towards long-term prosperity, the SDGs represent an important set of next steps in the evolution of transitions policy. While there is growing recognition regarding the potential for the SDGs to drive global-scale transformations towards more sustainable futures, the role of corporations in supporting the process and how the SDGs inform business models lacks clarity. Nevertheless, in many ways, the SDGs are tailored for companies looking to integrate sustainability into their business plans. The global SDG targets can be translated into a national context and framed to comply with national regulatory requirements while addressing sustainability specific to time and space [39]. Mawdsley [40] asserts that the private sector has particular strengths that can be utilized to deliver on the SDGs, which include but are not limited to a capacity for innovation, efficiency, responsiveness, and the provision of specific capabilities and resources.

Additionally, Martinuzzi et al. [41] suggest three ways the SDGs may prove beneficial as an underlying framework to guide corporate responsibility. First, the SDGs contain 17 agreed-upon sustainable development priorities broken down into targets of which many are directly relevant to business. Second, these globally accepted goals are endorsed by governments, businesses, and civil society, providing a common agenda for all stakeholders to rally around. Third, the SDGs fully

acknowledge the complexity, trade-offs, and systemic nature of sustainable development issues. Moving forward, the challenge for strategic CSR management is that of navigating a dynamic equilibrium, balancing short-term benefits with the long-term vision of sustainable development [19].

By aligning business approaches with the SDGs, corporate leaders can begin to redirect investment flows in a manner that maximizes value creation opportunities on sustainable development. Further, it can assist organizations in reducing risk, identifying opportunities, and determining long-term innovation solutions for addressing the SDGs. As a result, business and sustainable development agendas can and must be aligned if firms hope to move towards the macro-economic realities of sustained superior financial performance. This 'phenomenon-driven' review paper contributes important insights about the current state of research on SDGs and CSR and enriches the understanding of how the SDGs can drive the proliferation of strategic CSR. Situated within broader sustainability literature, this paper's concept of CSR is not static. Given the continual evolution of CSR as a standalone body of knowledge, a scoping review is warranted. Outlined by Tricco et al. [42], scoping reviews are different from systematic reviews and literature reviews in that the former provides a focus on diverse bodies of literature pertaining to a broad topic while the latter two direct search queries around a focused research question. Considering the ambitious fifteen-year agenda set forth by the SDGs and historical inconsistencies perpetuated by CSR research, this review provides a timely overview for clarifying key concepts while identifying the central implications of gaps and trends.

## 2. Materials and Methods

### 2.1. Scope of the Review

Scoping studies provide a grounded methodology for mapping concepts within a research domain and ontology of current research and practice-based evidence [43]. Past scoping reviews have been used by researchers to identify, organize, and analyze studies that are published within a domain to highlight knowledge gaps, develop future research agendas, and shed light on implications for decision-making [42,44]. A key component of the scoping process involves the clarification of reporting guidelines and stepwise checklists to ensure transparency, reliability, and repeatability of methods. This is particularly relevant given the strategic focus of this paper. Consequently, this review adopted the five-stage framework forwarded by Arksey and O'Malley [45], which includes: (1) identifying the scope of the study and research questions, (2) identifying the scale of relevant studies, (3) selecting relevant studies that match inclusion/exclusion criteria, (4) charting the data, and (5) summarizing and reporting the results.

While the reasoning behind why researchers might favor scoping reviews over more systematic counterparts varies, scoping reviews are viewed as a valid approach and an alternative when systematic review are not possible. According to Munn et al. (2018), scoping reviews are particularly "useful for examining emerging evidence when it is still unclear what other, more specific questions can be posed" [46] (p. 2). Given the recency and limited temporal period (i.e., 2015–2020) in which this study is focused, this review paper might be viewed as a natural precursor to future systematic reviews upon the conclusion of the 2030 agenda. By exploring 'whether' and 'how' the advent of the SDGs framework has impacted strategic CSR research, this paper might serve as a platform for informing future, more directed inquiries regarding the antecedents versus determinants and mediators versus moderators of this relationship.

### 2.2. Search Protocol

Using three scientific online databases (i.e., ISI Web of Science, Sci-Verse Scopus, and Science Direct), articles focused on corporate social responsibility and the Sustainable Development Goals were retrieved. These databases were chosen due to their broad coverage and advanced search capabilities, in order to maximize the inclusivity of the resulting dataset [47]. While we acknowledge the multiplicity



of terms used interchangeably with CSR throughout the literature, this paper emphasizes strategic CSR as expressed by Werther and Chandler [28] that denotes four key underpinnings:

1. Firms incorporate a CSR perspective within their strategic planning process;
2. Any actions firms take are directly related to core operations;
3. Firms incorporate a stakeholder perspective;
4. Firms shift from a short-term perspective to managing the firm's resources and relations with key stakeholders over the medium to long-term [28].

Given the specificity of this construct and the variability in which sustainability language materializes in practice, terms used synonymously were not included in the search of relevant publications. Electronic databases were searched, whereby articles containing the search term: "corporate social responsibl* AND sustainable development goal*" in their title, abstract, or keyword(s) were documented and stored using Mendeley reference management software. Designed to account for variance among suffixes and plural phrases, the same search term was used across databases to ensure consistency [47]. It is noted that scoping studies are capable of reporting evidence from a variety of sources including books, working reports, corporate disclosure documents, websites, rating agencies, and disclosure standards. However, given the limited time frame, this review focuses solely on journal articles under the assumption that most research output in this time period can be expected to have occurred in 'serial' periodicals. Moreover, and speaking to the nature of scoping reviews, this paper is selective and not exhaustive.

Following initial article retrieval and prior to any data filtration, a snowball approach was used to collect any unfound articles from our sample reference lists. This was conducted to increase the robustness of the article sample. The final step of sample refinement required that all retrieved articles meet one or more of the following inclusion/exclusion criteria [48]:

1. Articles must contain a direct reference to corporate social responsibility and the sustainable development goals in the title, keywords, or abstract;
2. CSR-related activities related to strategic firm-level initiatives were implemented in reference to the SDGs;
3. The impact of the SDGs on corporate operating models and relevance as an Integrated Reporting (IR) framework were discussed;
4. The content of the article met the definition and core underpinnings of strategic CSR set out by Werther and Chandler [28].

The initial search retrieved 146 peer-reviewed articles from the three databases between 2015 and 2020. Initial vetting and removal of duplicate articles left 91 articles remaining for consideration. After applying the inclusion/exclusion criteria, 56 articles were left to constitute the final data sample. A classification process of the SDGs mentioned per article was conducted using NVivo, which is qualitative data analysis software. Using the NVivo software provided data security and easy access and manipulation throughout the coding process (Creswell 2016). The final sample of 56 articles was analyzed using the following metrics: (1) author(s), (2) journal name, (3) year of publication, (4) publisher, (5) study location (by country), (6) SDGs covered, (7) sustainability dimensions (i.e., single, bi-combination, sustainability), and (8) sustainability research themes.

Previous studies were used to help inform and guide qualitative thematic coding processes and help saturate emergent criteria and elements. Coding is an iterative process of categorizing and sorting data, where codes represent categories that help summarize, synthesize, and organize themes characterizing a dataset [49]. Additionally, Gibb's [50] process of 'thematic' coding was applied to the final sample of this study due to its particular usefulness in creating codes that are analytically and theoretically robust rather than being purely descriptive. With the intent to deepen insights as to the proliferation of the SDGs in CSR research discourse, thematic coding emphasized the identification, analysis, and interpretation of patterns of meaning (or 'themes') within the dataset.

## 3. Research Findings and Results

### *3.1. Charting the Data*

The studies were published in reputable journals such as the Journal of Cleaner Production, which has the highest number of published articles on the subject, followed by the Sustainability journal, and finally the European Journal of Sustainable Development. For the full list of published articles per journal, see Appendix A Table A1.

### 3.1.1. Distribution of Studies Per Year

The analysis of this study shows that research on the topic of CSR and SDGs has increased substantially since 2015, with approximately 55% of the final sample being published in 2019 (see Figure 1). Given that the search protocol included articles published up to and including the end of January 2020, it is expected that the number of articles in 2020 is lower relative to previous years.

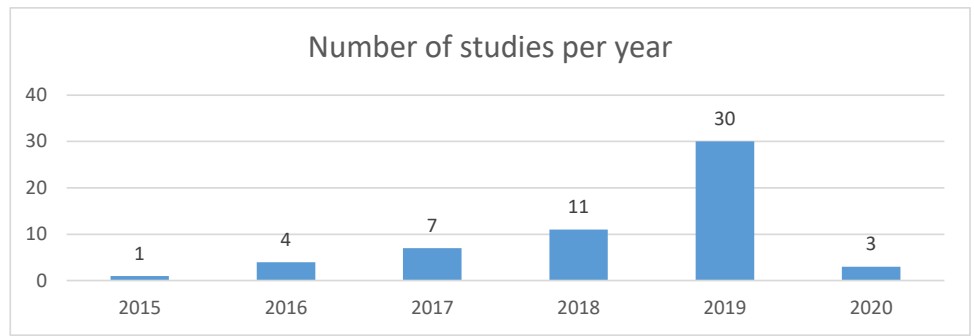

**Figure 1.** Number of studies per year.

### 3.1.2. Distribution of Studies by Country

The final sample is geographically diverse, including articles published in both developed and developing countries. As shown in Figure 2, the United Kingdom had the highest number of published articles followed by Australia, Spain, and Germany. Out of the 56 articles we analyzed in this scoping review, 48 articles were published in developed countries, while 8 publications were from developing countries. The degree to which geographic clustering can be expected to exist is largely dependent on stakeholder awareness and availability of slack resources, which currently favors markets in developed countries [19]. Noteworthy topics for future comparative analyses might focus on assessing geographical disparities outlining 'how' and 'why' strategic CSR varies across contexts, and measuring the depth and degree to which firms can realize the benefits of CSR engagement in developed versus developing economies. While controlling for organizational and contextual influences, the United Nations' SDGs framework should provide an internationally transferable measurement framework with 169 targets that might be translated and compared at the organizational level.

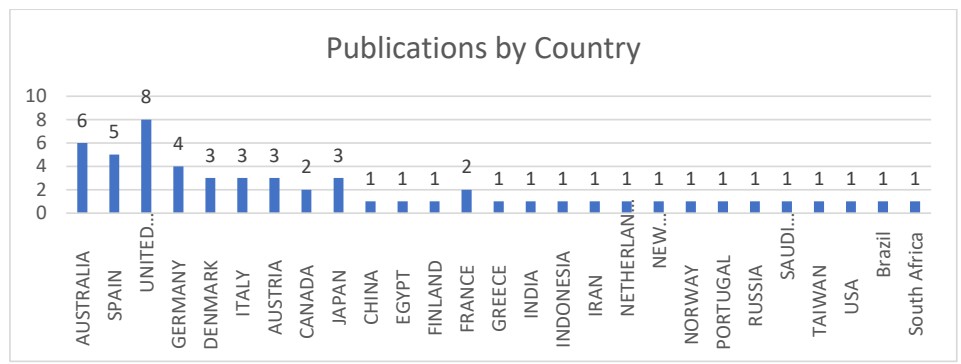

**Figure 2.** Publication by country.

### 3.1.3. Distribution of Articles Based on SDG Focused

As shown in Figure 3, a large proportion of articles were conducted under a generic lens, linking corporate CSR activities with a general mention of progress towards the achievement of the SDGs. Relatively, a smaller cohort of articles adopted a narrowed lens connecting specific SDGs to CSR activities. The following section of this paper provides a thematic analysis of the 56 articles and highlights the main SDGs within the papers, which are summarized in Table 1.

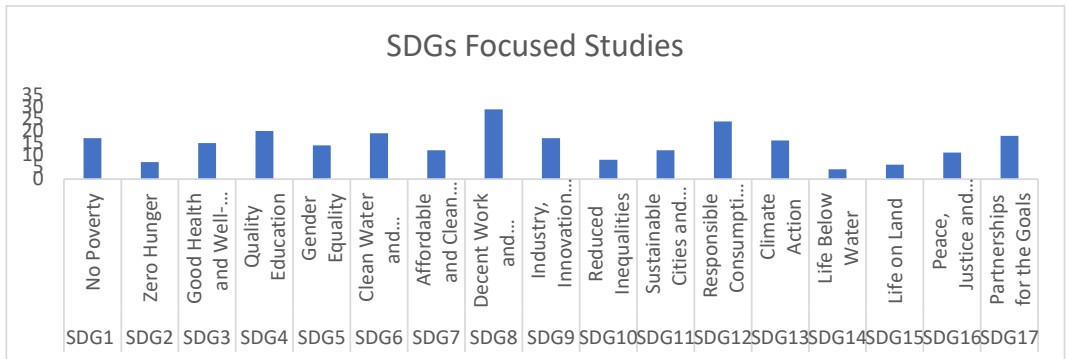

**Figure 3.** Distribution of focused articles based on Sustainable Development Goals (SDGs).

In line with findings of previous CSR research, the analysis of this study highlights the fact that there continues to be a hyper-emphasis on larger multi- and trans-national corporations in comparison to their small and medium enterprise counterparts [51].

### 3.2. Thematic Analysis

Using qualitative thematic coding methodology, a categorical framework for article classification was created. The content analysis approach was used to examine and assess the degree and nature of the influence of the SDGs on CSR literature. In this paper, the three categories of the sustainability dimensions framework by Alshehhi, Nobanee, and Khare [52] were adopted to analyze the distribution of the articles. The three categories are:

(1) Single-Dimension: Economic-Environmental-Social;
(2) Bi-Combination of dimensions: Socio-Economic, Economic-Environmental, and Social Environmental;
(3) Sustainability Dimension.

**Table 1.** Summary of Review Analysis.

| Source | Dimension | Strategic CSR | Research Focus | SDG(s) Covered |
|---|---|---|---|---|
| Naciti [70] | Socio-Environmental | ✔ | Company-focused | General |
| Poddar, Narula, and Zutshi [76] | Sustainability | ✔ | Company-focused | General |
| Grzeda [84] | Sustainability | ✕ | Company-focused | General |
| Contreras, Bos, and Kleimeier [60] | Economic | ✕ | Company-focused | General |
| Grover, Kar, and Ilavarasan [73] | Sustainability | ✔ | Company-focused | General |
| Calero, Garcia-Rodriguez De Guzman, Moraga, and Garcia [86] | Sustainability | ✕ | Company-focused | General |

**Table 1.** *Cont.*

| Source | Dimension | Strategic CSR | Research Focus | SDG(s) Covered |
|---|---|---|---|---|
| Cubilla-Montilla, Nieta-Librero, Galidno-Villardon, Vincente Galindo, and Garcia-Sanchez [57] | Social | ✗ | Community-focused | General |
| Fasoulis and Kurt [83] | Sustainability | ✔ | Company-focused | General |
| Buhmann, Jonsson, and Fisker [62] | Socio-Economic | ✗ | Company-focused | General |
| Perkiss, Dean, and Gibbons [87] | Sustainability | ✔ | Company-focused | General |
| Rosati and Fari [88] | Sustainability | ✔ | Company-focused | General |
| Barkemeyer and Miklian [85] | Socio-Economic | ✗ | Company-focused | SDG: 1, 8, 9, 12, 13 |
| Medina-Munoz and Medina-Munoz [53] | Social | ✗ | Company-focused | General |
| Denoncourt [74] | Sustainability | ✔ | Company-focused | SDG 9 |
| Lu, Ren, Lin, He, and Streimikis [89] | Sustainability | ✗ | Company-focused | General |
| Raj and Arun [90] | Sustainability | ✔ | Company-focused | General |
| Cantele and Zardini [51] | Sustainability | ✔ | Company-focused | General |
| Gunawan, Permatasari, and Tilt [91] | Sustainability | ✔ | Company-focused | General |
| Gider and Hamm [68] | Socio-Economic | ✗ | Consumer-focused | General |
| Sukhonos, Makarenko, Serpeninova, Drebot, and Okabe [92] | Sustainability | ✔ | Company-focused | General |
| Abdelhalim and Eldin [93] | Sustainability | ✔ | Company-focused | General |
| Munro and Arli [94] | Sustainability | ✔ | Company-focused | General |
| Stahl, Brewster, Collings, and Hajro [75] | Sustainability | ✔ | Company-focused | General |
| Liu [95] | Sustainability | ✔ | Company-focused | General |
| Zavyalova, Studenikin, and Starikova [54] | Social | ✔ | Company-focused | SDGs 1,3,4,5,6,8,10 |
| Miralles-Quiros, Miralles-Quiros, and Nogueira [67] | Socio-Economic | ✗ | Investor-focused | General |
| Avery and Hoope [61] | Economic | ✗ | Company-focused | General |
| Rahdari, Sepasi, and Moradi [64] | Socio-Economic | ✔ | Company-focused | General |
| Guandalini, Sun, and Zhou [96] | Sustainability | ✗ | Company-focused | General |
| Robinson, Martins, Solnet, and Baum [79] | Sustainability | ✗ | Employee-focused | SDG 8 |
| Avrampou, Skouloudis, Iliopoulos, and Khan [97] | Sustainability | ✗ | Company-focused | SDGs 8, 10, 12 |
| Rosati and Faria [88] | Sustainability | ✔ | Company-focused | General |
| Zimmermann [98] | Sustainability | ✔ | Company-focused | General |
| Berning [77] | Sustainability | ✗ | Company-focused | SDG 3,4, 8, 9,10, 11, 12, 13 |
| Kim [63] | Socio-Economic | ✗ | Company-focused | General |
| Manas-Viniegra [55] | Social | ✗ | Company-focused | General |

**Table 1.** *Cont.*

| Source | Dimension | Strategic CSR | Research Focus | SDG(s) Covered |
|---|---|---|---|---|
| Bosch-Badia, Montllor-Serrats, and Tarrazon-Rodon [81] | Sustainability | ✔ | Investor-focused | General |
| Calabrese, Costa, Ghiron, and Menichini [66] | Socio-Economic | ✕ | Employee-focused | SDG 5 |
| Yakovleva, Kotilainen, and Toivakka [99] | Sustainability | ✕ | Company-focused | General |
| Ekiugbo and Papanagnou [100] | Sustainability | ✕ | Company-focused | General |
| Wofford, MacDonald, and Rodehau [80] | sustainability | ✔ | Employee-focused | SDG 17, 3, 8 |
| Kelly [56] | Social | ✔ | Community-focused | General |
| Scheyvens, Banks, and Hughes [101] | Sustainability | ✔ | Company-focused | General |
| Sharma [102] | Sustainability | ✔ | Company-focused | General |
| Banik and Lin [103] | Sustainability | ✔ | Company-focused | SDG 8, 12 |
| Bull and Miklian [65] | Sustainability | ✔ | Company-focused | General |
| Soonsiripanichkul and Ngamcharoenmongkol [69] | Socio-Economic | ✕ | Consumer-focused | General |
| Nurunnabi, Esquer, Munguia, Zepeda, Perez and Velazquez [72] | Economic-Environmental | ✕ | Company-focused | SDG 7 |
| Selmier and Newenham-Kahindi [104] | Sustainability | ✕ | Company-focused | SDG 8,5,16 |
| Martinuzzi, Schonherr and Findler [105] | Sustainability | ✔ | Company-focused | General |
| Ramboarisata and Gendron [58] | Social | ✕ | Community-focused | General |
| Borges et al. [59] | Social | ✕ | Community-focused | SDG 4 |
| Naidoo and Gasparatos [71] | Economic-Environmental | ✕ | Company-focused | SDG 12 |
| Xia, Olanipekun, Chen, Xie, and Liu [106] | Sustainability | ✔ | Company-focused | General |
| Katamba [82] | Sustainability | ✔ | Company-focused | SDG 3 |
| Annan-Diab and Molinari [78] | Sustainability | ✔ | Community-focused | General |

Developed by the authors using articles retrieved from Scopus, Science Direct, and Web of Science.

### 3.2.1. Theme 1: Single-Dimension

The review of this study found nine articles that highlight a single dimension of CSR, specifically the social (seven studies) and economic dimension (two studies). These articles speak specifically of the social dimensions of corporate actions that aim at increasing societal welfare. As part of this paradigm, the role of internal and external stakeholders, along with specific institutions, are highlighted with respect to their role in driving CSR agendas toward achieving the SDGs. Specific sub-themes of corporate social action and performance include corporate contributions toward poverty alleviation [53], solutions to social issues [54], corporate CSR volunteering [55], and corporate-civil society partnerships [56]. Articles examining societal influence in driving CSR focus on cultural values as a normative institutional pressure [57] and the role of responsible management education [58,59].

Articles focusing on the economic dimension of CSR address sustainable finance and investment while elaborating on the centrality of the business-case of sustainability as a vector for continued CSR engagement. This includes Contreras et al. [60], who explore the drivers of adopting voluntary sustainability regulations in financial institutions. In addition, Avery and Hooper [61] studied how corporate CEOs can change organizational culture and performance by investing in CSR. Of the nine articles focused on the economic dimensions of CSR, only two (i.e., Kelly [56] and Zavyalova et al. [54]) discuss corporate responsibility from a strategic lens that views CSR as a strategic planning process that can only be achieved through partnerships among concerned stakeholders. Most articles associated with this theme explore the SDGs from a holistic approach, that being a general focus on the framework rather than a specific reference to one or more goals. Two notable exceptions include Zavyalova et al. [54] and Borges et al. [59]. The former article examines business projects that are aimed at solving social sustainability issues that can help achieve "socially-oriented" SDGs, specifically SDG 1, 3, 4, 5, 6, 8, and 10. The latter, Borges et al. [59], examine responsible management education hidden in the curriculum of business students with a focus on SDG 4, related to quality education.

### 3.2.2. Theme Two: Bi-Combination of Dimensions

The analysis highlights that some scholars tackle sustainability from a two-dimensional viewpoint, either (1) socio-economic, (2) socio-environmental, or (3) environmental-economic. In this review, eight articles examine CSR from a socio-economic dimension. In the first sub-category, namely the socio-economic dimension, the literature highlights the fact that organizations who invest in their CSR strategies should enhance their goodwill and develop trust from their stakeholders. Some authors adopted a corporate-oriented lens to reflect on the operationalization of CSR. For example, Buhmann et al. [62] explore how corporations can utilize their Human Resources (HR) towards achieving the SDGs. Likewise, Kim [63], Rahdari, Sepasi, and Moradi [64], and Bull and Miklian [65] analyze the socio-economic dimension of CSR from a corporate-driven standpoint, which highlights the positive economic and social gains for an organization to invest in CSR agendas. Calabrese, Costa, Ghiron, and Menichini [66] study the impact of gender equality on corporate governance, hence achieving robust CSR outcomes.

Nevertheless, some scholars focused on the socio-economic dimension of CSR from an outside-in approach, which targets external stakeholders such as investors [67] or customers [68,69]. The socio-economic articles all tackled the SDGs from a holistic perspective, except for Bull and Miklian [65], where the authors emphasize SDGs 1, 8, 12, and 13, which shed light on the economic and social implications of businesses.

Additionally, in the same category of the two-dimensional CSR strategies are the socio-environmental and the economic-environmental perspectives. In this review, only one article, that being Naciti [70], uses a socio-environmental lens to examine the role of an institution's Board of Directors in achieving better sustainability performance with a higher prominence on the social and environmental pillars. The author uses a strategic CSR framework that highlights the long-term dimension of CSR, which necessitates strategic collaboration among concerned stakeholders. The author uses a company-focused viewpoint with a holistic overview of the 17 SDGs. Finally, the economic-environmental sub-category included two articles. The first, by Naidoo and Gasparatos [71], examines the sustainability drivers within CSR agendas as well as the performance measurement and reporting in corporations. This article focuses on SDG 12 and identifies best practices for responsible consumption and production in the SDGs era. Likewise, Nurunnabi et al. [72] analyze energy efficiency as a tool to achieve the SDGs with a specific focus on SDG 7.

### 3.2.3. Theme 3: Sustainability Dimension Studies

In the last categorization of this review, we identified articles that study CSR from a comprehensive viewpoint that covers the economic, social, and environmental dimensions of sustainability. Out of the 56 articles included in this scoping review, 36 articles analyzed CSR from a comprehensive approach

that aims to balance the economic, social, and environmental pillars of sustainability. The majority of these articles (32 articles) have a company-focused approach, such as exploring the impact of CSR on company reputation [73], identifying products, and process innovation, within organizations towards achieving the SDGs [74]. The research on large organizations and multinationals still dominate the literature on CSR [75–77], with little emphasis on the role of small and medium enterprises in achieving the SDGs through their CSR agendas.

Moreover, some scholars in the sustainability dimension used a community-focused lens to highlight the needs of interdisciplinary education programs in the academic world and industry to help achieve the SDG via strategic CSR approaches [78]. Other scholars adopted an employee-focused lens that highlights the importance of decent working conditions for employees [79], especially gender issues in the workplace [80]. Finally, some used an investor-focused lens that explores the role of responsible investors in achieving the SDGs [81]. The majority of the articles in this theme (24 articles) covered SDGs in a generic sense. Yet, studies such as Denoncourt [74], Katamba [82], and Robinson et al. [79] tried to link specific goals with the CSR practices of companies such as SDGs 8, 12, and 13.

### 3.3. Summary of Scoping Review Results

Table 1 summarizes the results of the scoping review. Although some single- and bi-dimensional articles exploring CSR from one- or two-dimension(s) view CSR as a strategic planning process, articles adopting a comprehensive approach to CSR are the main articles tackling CSR from a strategic lens such as Poddar, Narula, and Zutshi [76], Grover, Kar, and Ilavarasan [73], and Fasoulis and Kurt [83].

From a research-focussed perspective, the articles under review were classified according to whether their studies focused on companies or other internal or external stakeholders such as employees, consumers, investors, and the wider community. The analysis of this study shows that most articles that follow a comprehensive sustainability approach are company focused. A limited number of articles tackle sustainability from a stakeholder perspective, for example, Gider and Hamm [68], Miralles-Quiros, Miralles-Quiros, and Nogueira [67], and Wofford, MacDonald, and Rodehau [80].

Finally, the majority of the articles under study discuss CSR in relation to SDGs in a generic manner such as Naciti [70], Grzeda [84], and Buhmann, Jonsson, and Fisker [62]. On the other hand, some studies tackled specific SDGs in their studies. For instance, Denoncourt [74] examined the connection between CSR and SDG 9, "industry, innovation, and infrastructure". Likewise, Calabrese, Costa, Ghiron, and Menichini [66] specifically studied the presence of SDG 5, "gender equality" among CSR managers. Other articles, however, mentioned more than one SDG in their studies. For instance, Barkemeyer and Miklian [85] explored the implications of their results on more than one SDG, and Zavyalova, Studenikin, and Starikova [54] attempted to frame the CSR initiatives of a leading multinational company under the umbrella of a number of SDGs. Overall, this review opens various potential avenues for new research in the business-society field specifically, and in the sustainable development discipline in general. Future research recommendations are discussed in the following section.

## 4. Discussion and Implications for Future Research

The SDGs offer a shared vision; a roadmap by which businesses can begin to strategically align firm-level CSR initiatives with both national and international sustainable development agendas. This CSR-SDG nexus is crucial in enhancing the contribution of CSR to sustainable development. Based on the review conducted in this study, future research insights for more strategic implementations of CSR that can effectively contribute to the successful achievement of the development goals are highlighted below:

### 4.1. Investigating Actual Corporate Contribution to Sustainable Development

An emergent theme in this analysis highlights the role of corporations in elevating social problems and enhancing the well-being of society. While companies increasingly take action to improve

their social and environmental performance, the effectiveness of these efforts in advancing progress toward the SDGs remains poorly understood. This points towards a critical gap and a lingering need for empirically grounded research and evidence-based management systems that are necessary to accelerate and scale up the adoption of governance structures, reporting methods, and management innovations to achieve sustainable development. More specifically, deepened understandings of the impact(s) of CSR activities on sustainable development and the achievement of the SDGs is required. For instance, future research can assess the impact of a specific business sector as a whole or comparatively study the impact of one sector on other business sectors [71].

In addition, the effectiveness of various CSR initiatives emphasizing poverty alleviation and the underlying driving forces of implementation can be explored. In this vein, corporations can enhance the strategic integration of social SDGs, specifically those related to poverty alleviation, by conducting and reporting short-term social targets and communicating them across the company [56]. Such corporate dedication towards a specific goal can eventually lead to the promotion of other SDGs (i.e., positive spillover). As a vehicle for more precipitous change, action of this nature holds the potential to increase the scale and rate at which CSR-related impacts address macro-level sustainable development problems [54]. Moving forward, future research might begin to explore the most effective strategies companies can adopt to implement CSR projects that can effectively contribute to the long-term sustainable development of their societies.

While research exists on the role of multinational companies (MNCs) in driving sustainability, there is a need to further investigate the actual, and absolute, positive contribution(s) of an MNC's CSR practices on sustainable development. Particularly, how does and what are the absolute impacts of firm-level CSR initiatives on the degree to which a given MNC is successful in attaining a given SDG. The notion of "giving back to society" in a mere philanthropic sense is becoming less legitimate in the eyes of stakeholders. As a result, this 'business as usual' approach has left the ability of businesses to face sustainable development challenges in question [41]. Impact assessment studies should be performed to identify the positive developments that MNCs enhanced and the negative challenges that MNCs were able to eradicate or reduce. Because the SDGs highlight a wide range of sustainable development issues, more research is needed to identify the main Sustainable Development Goals where MNCs can provide the most significant positive impacts in various contexts and across different industries. Overall, it is important to gain insight into how the SDGs are being integrated into strategy and understand the potential value creation provided through the SDG framework. This will help research move towards more accurate and precise evaluations when determining the extent to which core CSR initiatives strategically address sustainable development problems and effectively implement the goals.

*4.2. The Role of Financial Institutions in Driving Sustainable Development*

While there is increasing interest in the role financial institutions play in driving sustainable development [60], more research is required to measure the impacts of institutional action and the degree of influence they have on sustainable development progress. For instance, future studies might elaborate on the role of financial institutions in financing SDGs. This represents a significant element in enabling collective action and ensuring continued sub-national progress on relevant SDGs. In addition, future research might examine the triple bottom line impacts of adopting voluntary social and environmental guidelines (i.e., the Principles for Positive Impact Finance and the Equator Principle) on the degree and nature of cooperative collaboration between financial institutions, businesses, and civil society members, specifically non-government organizations (NGOs). Weber and Feltmate [4] highlight the fact that voluntary self-regulatory principles, specifically the Equator Principle, significantly increase cooperation between institutions that adopt them. This draws specific attention to SDG 17, Partnerships for the Goals, and its centrality to enabling symbiotic, multi-stakeholder action networks. Exploring how such cooperation can contribute to the strategic implementation of CSR and the achievement of the SDGs amongst adopting institutions is an important avenue for future research.

*4.3. The Role of Stakeholders in the Implementation of Strategic CSR Models*

More attention should be given to the role of stakeholders in achieving effective implementation of the goals across the globe. It is argued that the SDGs are "macro-ethics" where individual ethics represent a main building block of its achievement and success [81]. A likely research question can examine the ethical behavior of individuals that highly support or discourage the path to sustainable development. In addition, future research can investigate the ethical awareness of individuals concerning sustainability and whether it can positively impact the attainment of the goals. On the other hand, the role of the government as a powerful stakeholder that can drive strategic CSR should not be ignored in future research. Legislative frameworks on CSR can provide incentives to the private sector to implement strategic CSR models and could act as a shield against declining CSR practices, specifically in unfavorable economic conditions. Yet, the government alone is not enough. The successful embedment of the SDGs in the CSR practices of the private sector cannot be achieved without the active involvement of all stakeholders [93]. Accordingly, future research should pay closer attention to how companies can utilize and empower stakeholders, such as social entrepreneurs, to further enhance the strategic outcome of CSR activities in realizing SDG objectives on both the micro, meso, and macro scales [64].

*4.4. Increasing Research on Sustainability in Under-Researched Areas*

The subject of sustainability in small and medium enterprises has been largely ignored in the examined literature. For the SDGs to be successfully achieved, the role of small and medium businesses should be considered [51]. It is an unprecedented time where a universal sustainable development vision exists where businesses are considered significant partners in shaping their success [41]. Therefore, there is a need to widen the scope of CSR literature in the SDG era to encompass, not only large corporations, but also small and medium enterprises (SMEs). Future research is needed on the potential contribution of SMEs towards the achievement of the SDGs and on the impacts of SMEs' social and environmental activities.

Finally, while developing countries require "inclusive business models" that link CSR strategies with sustainable development and the SDGs, studies that explicitly tackle a strategic approach to CSR in the context of developing countries are outnumbered/very few. Understanding the impact of the CSR practices of the private sector in developing countries and to what extent they are aligned with the SDGs is crucial to facilitate the achievement of the goals and help solve the major social and environmental challenges faced by such countries. Future research is needed on the impact assessment of CSR practices, both quantitatively and qualitatively, to strategically develop CSR in developing countries and redirect it towards the attainment of the goals [93].

## 5. Conclusions

The global adoption of the United Nations' Sustainable Development Goals has shifted what society should expect of companies in their communities and their role as leaders in the global sustainability transition. Nevertheless, it should be clear that organizations are still among the largest contributors to issues of sustainability. Despite known profitability and mounting evidence pointing to the multiplicity of direct and indirect benefits, there remains a lingering reluctance to undertake strategic CSR initiatives. The field of management sciences has made notable strides in supporting corporate transitions toward more sustainable futures, led by an ambitious action-oriented research agenda. Emerging as a functional response to the innate difficulties in managing the sustainability paradox and making progress on sustainability targets, management sciences, as a field of research, holds the potential to ground practitioner decision-making processes in empirical evidence. While it is acknowledged that this scoping review falls short in establishing causality between the evolution of CSR research trends and progress on the SDGs, this paper should serve as an entry point for future scientific inquiry.

With the intent to contribute to evidence-based management literature on CSR, this review advocates the development of a Global Ontological Framework for CSR practices. By deepening insights regarding the nature of diffusion, depth of integration, and degree of influence of the SDGs into CSR practices, this review highlights key areas of overlap between research and macro-level policy discourse. Future research in managerial sciences may use the findings of this review to explore potential causality and bi-directional influences between specific SDGs and strategic CSR research. Such findings could inform corporate strategy, operations, product development, and supply chain management in a manner that would minimize the impacts of cognitive biases latent to managerial decision-making and rational limitations to corporate governance structures.

The results of this review provide the initial framings of a roadmap in regards to changing expectations of corporate responsibility in the SDG era, which SDGs are influencing corporate strategy and CSR agendas, how the SDGs are affecting stakeholder expectations and regulatory requirements, and provides examples of how the SDGs are being integrated into CSR reporting. By presenting a summary of current research on SDGs and CSR while highlighting under-researched and unexplored research areas, more knowledge for the advancement of this field can be gained. Further, based on the suggested areas for future research, theoretical perspectives related to how the SDGs inform sustainable business models and the role of stakeholders in promoting corporate commitment to sustainable development can be extended.

Peer-reviewed articles on CSR and the SDGs have increased substantially over the study period, specifically in 2019. Over half the articles published since 2015 were published in 2019, and early indications based on the first two months of 2020 suggest the trend will continue. While the authors of this paper acknowledge the importance of all 17 SDGs, this study indicates that, to date, research has placed a particular emphasis on SDGs 1, 4, 6, 8, 12, and 17. Given the voluntary nature and lack of a single ubiquitous reporting standard spanning geographic contexts and industries, the capacity for CSR to positively drive the global sustainability agenda remains impeded. While this review is descriptive in its analysis of CSR-SDG linkages, future systematic reviews might seek to quantify the relationships highlighted by this study. Moreover, the degree and depth of CSR-SDG interactions might be impacted by geographic context, and as such, warrants future investigation to comparatively assess developing countries and not only developed ones. Nevertheless, it is the opinion of this paper that the UN SDGs represent a touchstone that offers a unifying framework for corporate reporting. Moving forward, this will require continuous efforts to ensure comparability and consistency in reporting mechanisms that create functional linkages between: (1) national and global commitments to sustainable development; and (2) corporate and civil society actions to building sustainable and resilient communities. Integrated reporting is particularly helpful for increasing the understanding of the importance of sustainable development issues to value creation because of its multi-capital approach, long-term focus, guiding principle of connectivity, and a requirement for board involvement.

In this paper, several important areas of future research have been identified to understand the role of corporations in supporting the societal achievement of the SDGs by 2030. First, current literature focuses mainly on the role of large organizations, yet further analysis is needed to explore the SDGs as a framework to inform the activities and performance of small and medium sized enterprises. Second, further studies might consider analyzing the role of governments in advancing strategic CSR practices. Finally, future research might consider examining the role of stakeholders in driving strategic CSR including employees, investors, consumers, and civil society members. Overall, the findings of this study affirm the need to understand the environmental and social impacts of CSR activities on sustainable development and how current CSR performance can be improved and redirected to have long-term sustainable benefits on companies and society at large.

The limitations of this study are twofold. First, the selection process mainly focused on pre-defined keywords, which may have resulted in the exclusion of relevant papers. However, the study used three scientific online databases (i.e., ISI Web of Science, Sci-Verse Scopus, and Science Direct) to obtain studies on CSR in the SDG era covering a period of more than five years (2015–2020). This wide-scope

analysis can provide extensive evidence on the interconnection between CSR and sustainability goals. Second, the findings of the study on CSR activities are mainly derived from academic journal articles rather than industry sources, such as corporate websites and corporate sustainability reports. Yet, the majority of the examined articles in this study are all applied research covering a wide range of industry sectors, which may be sufficient to reveal the status quo of CSR in a post SDG era. Moving forward, global policy studies must give greater emphasis to understanding processes of actual organizational change in CSR practices over time. This review provides a timely response highlighting the continued evolution of CSR research in a post-SDGs era. However, if the UN SDGs are to assume their role as a universally accepted corporate reporting standard, future research will require clarity and causality regarding 'how' and 'to what degree' interaction(s) between mediating mechanisms, context, and outputs influences the nature of CSR-SDG relationships.

**Author Contributions:** Conceptualization, A.E., N.P., J.W.; data curation, A.E., N.P.; formal analysis, A.E., D.E.-B.; methodology, A.E., N.P., D.E.-B.; software, D.E.-B.; supervision, O.W., J.W.; writing—original draft, A.E., N.P., J.W., D.E.-B.; writing—Review and Editing, O.W. All authors have read and agreed to the published version of the manuscript.

**Funding:** This research received no external funding.

**Conflicts of Interest:** The authors declare no conflict of interest.

## Appendix A

**Table A1.** Full list of the number of published articles per journal.

| Journal | Published Articles |
|---|---|
| JOURNAL OF CLEANER PRODUCTION | 6 |
| SUSTAINABILITY | 5 |
| EUROPEAN JOURNAL OF SUSTAINABLE DEVELOPMENT | 3 |
| CORPORATE SOCIAL RESPONSIBILITY AND ENVIRONMENTAL MANAGEMENT | 2 |
| JOURNAL CORPORATE SOCIAL RESPONSIBILITY AND ENVIRONMENTAL MANAGEMENT | 2 |
| SUSTAINABLE DEVELOPMENT | 2 |
| TRANSNATIONAL CORPORATIONS | 2 |
| BUSINESS AND POLITICS | 1 |
| BUSINESS ETHICS | 1 |
| BUSINESS STRATEGY AND THE ENVIRONMENT | 1 |
| CENTRAL ASIA AND THE CAUCASUS | 1 |
| CORPORATE GOVERNANCE (BINGLEY) | 1 |
| EXTRACTIVE INDUSTRIES AND SOCIETY | 1 |
| GEO JOURNAL | 1 |
| GLOBALIZATION AND HEALTH | 1 |
| HUMAN RESOURCE MANAGEMENT REVIEW | 1 |
| INTERNATIONAL JOURNAL OF INFORMATION MANAGEMENT | 1 |
| INTERNATIONAL JOURNAL OF NONPROFIT AND VOLUNTARY SECTOR MARKETING | 1 |
| JOURNAL OF CORPORATE LAW STUDIES | 1 |
| JOURNAL OF MANAGEMENT SPIRITUALITY AND RELIGION | 1 |
| JOURNAL OF SUSTAINABLE TOURISM | 1 |
| JOURNAL BUSINESS AND POLITICS | 1 |
| JOURNAL INTERNATIONAL FOOD AND AGRIBUSINESS MANAGEMENT REVIEW | 1 |
| JOURNAL INTERNATIONAL JOURNAL OF SOCIOLOGY AND SOCIAL POLICY | 1 |

**Table A1.** *Cont.*

| Journal | Published Articles |
| --- | --- |
| JOURNAL INTERNATIONAL JOURNAL OF SUSTAINABLE DEVELOPMENT AND WORLD ECOLOGY | 1 |
| JOURNAL OF HUMAN RIGHTS PRACTICE | 1 |
| WASHINGTON INTERNATIONAL LAW JOURNAL | 1 |
| PROBLEMS AND PERSPECTIVES IN MANAGEMENT | 1 |
| SOCIAL AND ENVIRONMENTAL ACCOUNTABILITY JOURNAL | 1 |
| SOCIAL SCIENCES | 1 |
| STRATEGY AND LEADERSHIP | 1 |
| WORLD DEVELOPMENT | 1 |

Developed by the authors using articles retrieved from Scopus, Science Direct, and Web of Science.

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
