# Peer review of "Scoping the Evolution of Corporate Social Responsibility (CSR) Research in the Sustainable Development Goals (SDGs) Era"

_sustainability, doi:10.3390/su12145544_

Round 1
Reviewer 1 Report
The article is very well documented, very well structured and proposes an interesting topic to the readers of the journal.
My recommendations address the following issues
1. The manuscript must be written in objective third person point of view throughout (Use "the authors" or "the researchers" NOT "I" or "we").
2. All tables must present the source of the information presented.
3. I recommend the authors to remind the unethical attitude of some companies that initiate CSR programs only as a marketing strategy, for image improvement (for example, greenwashing strategy )
a) Nguyen, T. T. H., Yang, Z., Nguyen, N., Johnson, L. W., & Cao, T. K. (2019). Greenwash and green purchase intention: The mediating role of green skepticism. Sustainability, 11(9), 2653.
b) Pimonenko, T., Bilan, Y., Horák, J., Starchenko, L., & Gajda, W. (2020). Green Brand of Companies and Greenwashing under Sustainable Development Goals. Sustainability, 12(4), 1679.
c) Urbański, M. (2020). Are You Environmentally Conscious Enough to Differentiate between Greenwashed and Sustainable Items? A Global Consumers Perspective. Sustainability, 12(5), 1786.
Author Response
Dear Reviewer,
Thank you for your thoughtful review and constructive comments that improve the quality of the paper. We addressed all the comments as noted below. By considering your suggestions it has helped strengthen the entire paper and clarify several aspects. We appreciate your time and support.
Sincerely
The authors
Response to Reviewer #1:
The article is very well documented, very well structured and proposes an interesting topic to the readers of the journal.
Thank you for your comment we appreciate it and hopefully, the revised version meets your expectations.
(1) The manuscript must be written in objective third person point of view throughout (Use "the authors" or "the researchers" NOT "I" or "we")
Thank you. The manuscript has been revised and written objectively. Words such as “I” and “we” were removed, and the sentences were restructured and edited.
(2) All tables must present the source of the information presented.
Thank you. We have added the source of information based on the databases used: Scopus, Science Direct, and Web of Science.
(3) I recommend the authors to remind the unethical attitude of some companies that initiate CSR programs only as a marketing strategy, for image improvement (for example, greenwashing strategy)
- a) Nguyen, T. T. H., Yang, Z., Nguyen, N., Johnson, L. W., & Cao, T. K. (2019). Greenwash and green purchase intention: The mediating role of green skepticism. Sustainability, 11(9), 2653.
- b) Pimonenko, T.,Bilan, Y.,Horák, J., Starchenko, L., & Gajda, W. (2020). Green Brand of Companies and Greenwashing under Sustainable Development Goals. Sustainability, 12(4), 1679.
- c)Urbański, M. (2020). Are You Environmentally Conscious Enough to Differentiate between Greenwashed and Sustainable Items? A Global Consumers Perspective. Sustainability, 12(5), 1786.
Thank you. Several paragraphs have been added to address this recommendation in full. Content is copied below.
Whether pursuing business cases of CSR is enough to satisfy global sustainable development remains subject to debate within and across academic disciplines. Often resting on a priori organizational frameworks, the legitimacy of this logic falls short when sustainable development is reduced under neo-liberal economic rationality or economic performance leveraged with coincidental CSR contributions [17]. In practice, bottom-line implications are left vulnerable to capricious public opinions, senior management turnover, and quarterly financial cycles [18]. Deeply ingrained throughout conventional cost accounting and performance management is a utilitarian view that rewards manager’s and senior leadership when acting as self-seeking opportunistic individuals with the intent to maximize personal economic interests [19]. Materializing in the form of ‘greenwashing’, the reduction of CSR under win-win scenarios at the intersection of the triple bottom long constitutes a key managerial motivation for CSR and a conventional approach to building the business case [20]. Rather than an end in-of-itself, CSR activities are treated as philanthropic add-ons necessary for catering to current public opinion while securing loyalty [17, 21]. This does little in the way of transforming organizational behaviour in a manner that is required to support meaningful progress on the SDGs. This underscores that the very notion of ‘doing well by doing good’ is fundamentally a proposition of diminishing returns [22, 23].
Demonstrable of a lack of managerial know-how and information for intervention selection/design, research indicates that such realities negatively mediate management’s motivation/ commitment to CSR [12]. Until this rationale is addressed systematically, strategic CSR literature will continue to turn-out isolated success stories. As identified by Schaltegger et al. (2012), this will require that the formulation and implementation of strategy moves away from those that only strive for market sustainability through competitive advantages in the sense of the Resource-Based View (RBV) of the firm. By aligning the purpose and values of CSR with market drivers and stakeholder demands, CSR practices have become due diligence for preserving the firm's license to operate, avoiding reputational damages, building loyalty, and maintaining competitive positioning [9]. Empirically grounded, the impacts of CSR on financial performance can be explained through top-line growth [7], decreased cost to capital, increased reputation and goodwill [8], and reduced technical and material risks [9].
With respect to research, Bansal and Song [24] highlight that despite novel insights being made on the role of the firm and its embeddedness within the business-society-nature interface, the variability among its subjective interpretations has limited construct validity in practice. Nevertheless, since the introduction of the SDGs, many firms have begun to strategically engage with the international framework as a means of creating functional linkages between performance outcomes and the common good [25]. An integrated framework comprised of 169 targets and 232 unique indicators, the SDGs have shifted CSR discourse from being reactive to stakeholders’ mandates to a proactive one that helps firms play an active role in influencing sustainable development trajectories [26].
We want to thank you again for the highly valuable work that you have put into the review of our paper and for the support of our research. We hope that we have been able to justify your effort through the changes we have made in response to all your comments.
Reviewer 2 Report
The article is very thought-provoking because it organizes published researches on the relationship between Sustainable Development Goals (SGD) and Corporate Social Responsible (CSR). The considerations are based on current literature, the research method was selected correctly, and the research itself was also carried out in accordance with the adopted methodology.
However, I want to draw attention to two issues that I believe need to be supplemented. The first refers to the title of the article, which begins with the words: Redefining Responsibility? There is no explicit answer to the somewhat rhetorical question: in the context of the analyzed literature, how should responsibility be redefined today?
My second doubt is related to the contribution of conclusions to the development of management sciences. What the presented analysis brings new to management sciences. I feel the lack of a clear summary of the contribution of these considerations to the development of management sciences of contemporary organizations.
Author Response
Dear Reviewer,
Thank you for your thoughtful review and constructive comments that improve the quality of the paper. We addressed all the comments as noted below. By considering your suggestions it has helped strengthen the entire paper and clarify several aspects. We appreciate your time and support.
Sincerely,
The authors
- The article is very thought-provoking because it organizes published researcheson the relationship between Sustainable Development Goals (SGD) and Corporate Social Responsible (CSR). The considerations are based on current literature, the research method was selected correctly, and the research itself was also carried out in accordance with the adopted methodology.
Thank you for your comment we appreciate it and hopefully, the revised version meets your expectations.
- However, I want to draw attention to two issues that I believe need to be supplemented. The first refers to the title of the article, which begins with the words: Redefining Responsibility? There is no explicit answer to the somewhat rhetorical question: in the context of the analyzed literature, how should responsibility be redefined today?
Thank you. The title of the paper has been modified to better reflect the purpose of the paper. We changed the title to:
Scoping the evolution of Corporate Social Responsibility (CSR) Research in the Sustainable Development Goals (SDGs) World
- My second doubt is related to the contribution of conclusions to the development of management sciences. What the presented analysis brings new to management sciences. I feel the lack of a clear summary of the contribution of these considerations to the development of management sciences of contemporary organizations
Thank you kindly for your insight. We have revisited and integrated as deemed appropriate. Copied below is the newly integrated content connecting our findings to the field of management sciences.
The global adoption of the United Nations’ Sustainable Development Goals has shifted what society should expect of companies in their communities and their role as leaders in the global sustainability transition. Nevertheless, it should be clear that organizations are still among the largest contributors to issues of sustainability. Despite known profitability and mounting evidence pointing to the multiplicity of direct and indirect benefits, there remains a lingering reluctance to undertake strategic CSR initiatives. The field of Management Sciences has made notable strides in supporting corporate transitions toward more sustainable futures led by an ambitious action-oriented research agenda. Emerging as a functional response to the innate difficulties in managing the sustainability paradox and making progress on sustainability targets, management sciences, as a field of research, holds potential to ground practitioner decision-making processes in empirical evidence. While it is acknowledged that this scoping review falls short in establishing causality between the evolution of CSR research trends and progress on the SDGs, this paper should serve as an entry point for future scientific inquiry.
With the intent to contribute to evidence-based management literature on CSR, this review advocates for the development of a Global Ontological Framework for CSR practices. By deepening insights regarding the nature of diffusion, depth of integration, and degree of influence of the SDGs into CSR practices, this review highlights key areas of overlap between research and macro-level policy discourse. Future research in managerial sciences may use the findings of this review to explore potential causality and bi-directional influences between specific SDGs and strategic CSR research. Such findings could inform corporate strategy, operations, product development, and supply chain management in a manner that would minimize the impacts of cognitive biases latent to managerial decision-making and rational limitations to corporate governance structures.
The results of this review provide the initial framings of a roadmap in regards to changing expectations of corporate responsibility in the SDG era; which SDGs are influencing corporate strategy, and CSR agendas; how the SDGs are affecting stakeholder expectations and regulatory requirements; and examples of how the SDGs are being integrated into CSR reporting. By presenting a summary of current research on SDGs and CSR while highlighting under-researched and unexplored research areas, more knowledge for the advancement of this field can be gained. Further, based on the suggested areas for future research, theoretical perspectives related to how the SDGs inform sustainable business models and the role of stakeholders in promoting corporate commitment to sustainable development can be extended.
We want to thank you again for the highly valuable work that you have put into the review of our paper and for the support of our research. We hope that we have been able to justify your effort through the changes we have made in response to all your comments.